# Clinical (BMI and MRI) and Biochemical (Adiponectin, Leptin, TNF-α, and IL-6) Effects of High-Intensity Aerobic Training with High-Protein Diet in Children with Obesity Following COVID-19 Infection

**DOI:** 10.3390/ijerph19127194

**Published:** 2022-06-11

**Authors:** Gopal Nambi, Mshari Alghadier, Tamer E. Elnegamy, Reem M. Basuodan, Reem M. Alwhaibi, Arul Vellaiyan, Naif A. Nwihadh, Osama R. Aldhafian, Anju Verma, Shahul Hameed Pakkir Mohamed, Mohamed Faisal Chevidikunnan, Fayaz Khan

**Affiliations:** 1Department of Health and Rehabilitation Sciences, College of Applied Medical Sciences, Prince Sattam bin Abdulaziz University, Al-Kharj 11947, Saudi Arabia; m.alghadier@psau.edu.sa (M.A.); tameremam22@gmail.com (T.E.E.); 2Department of Rehabilitation Sciences, College of Health and Rehabilitation Sciences, Princess Nourah bint Abdulrahman University, P.O. Box 84428, Riyadh 11671, Saudi Arabia; rmbasoudan@pnu.edu.sa (R.M.B.); rmalwhaibi@pnu.edu.sa (R.M.A.); 3Department of Nursing, College of Applied Medical Sciences, Prince Sattam bin Abdulaziz University, Al-Kharj 11947, Saudi Arabia; a.vellaiyan@psau.edu.sa; 4Department of Surgery, College of Medicine, Prince Sattam bin Abdulaziz University, Al-Kharj 11947, Saudi Arabia; n.alniwhadh@psau.edu.sa (N.A.N.); o.aldhafian@psau.edu.sa (O.R.A.); 5Department of Exercise and Sports, University of Sydney, Sydney, NSW 2006, Australia; aver8191@uni.sydney.edu.au; 6Department of Physical Therapy, Faculty of Applied Medical Sciences, University of Tabuk, Tabuk 71491, Saudi Arabia; s-mohamed@ut.edu.sa; 7Department of Physical Therapy, Faculty of Medical Rehabilitation Sciences, King Abdulaziz University, Jeddah 22254, Saudi Arabia; mfaisal@kau.edu.sa (M.F.C.); fayazrkhan@gmail.com (F.K.)

**Keywords:** aerobic training, adipocytokines, children with obesity, COVID-19, high protein diet

## Abstract

Objective: To find the clinical and biochemical effects of high-intensity aerobic training with a high-protein diet in children with obesity following COVID-19 infection. Methods: By using the block randomization method, the eligible participants were randomized into two groups. The first group received high-intensity aerobic training with a high-protein diet (Group A; *n* = 38) and the second group were allowed to do regular physical activities and eat a regular diet (Group B; *n* = 38) for 8 weeks. Clinical (basal metabolic index (BMI) and muscle-mass-cross-sectional area (CSA)) and biochemical (Adiponectin, leptin, TNF-α, and IL-6) measures were measured at baseline, on the 8th week, and at 6-months follow-up. Results: Baseline demographic and clinical attributes show homogenous presentation among the study groups (*p* > 0.05). After eight weeks of intervention, and at the end of 6-months follow-up, the basal metabolic index (BMI) (6.3) (CI 95% 4.71 to 7.88), mid-arm CSA (17) (CI 95% 14.70 to 19.29), mid-thigh CSA (13.10) (CI 95% 10.60 to 15.59), mid-calf CSA (11.3) (CI 95% 9.30 to 13.29), adiponectin (−1.9) (CI 95% −2.13 to −1.66), leptin (5.64) (CI 95% 5.50 to 5.77), TNF-α (0.5) (CI 95% 0.42 to 0.57), and IL-6 (0.21) (CI 95% 0.18 to 0.23) showed more improvement (*p* < 0.001) in Group A than Group B (*p* > 0.05). Conclusion: Overall, this trial found that high-intensity aerobic training with a high-protein diet decreased the BMI percentile and muscle mass (arm, thigh, and calf), and positively altered the biochemical variables in children with obesity.

## 1. Introduction

The coronavirus disease (COVID-19) is a communicable pandemic disease, as stated by the World Health Organization (WHO), which has been affecting the world since December 2019. It is not only a major health concern to the patient, but also creates financial and community issues, and 5% of children between the age of 5 and 12 are affected by it [1]. The coronavirus enters into the body via the nasopharyngeal route and gradually affects all systems of the body. The spikes of the coronavirus contain a special protein, which attaches to the receptor present on the cell membrane of the host cell, and enters into the cell and deactivates it [2]. COVID-19-infected children develop the signs and symptoms of the disease, which can be exaggerated or life threatening when associated with comorbidities such as obesity, sickle cell anemia, immune disorders, chromosomal abnormalities, chronic respiratory or cardiac problems, and congenital malformations. It has been observed that obese children affected with COVID-19 who are physically inactive or in a sedentary lifestyle may exacerbate the condition [3]. It is a major health concern in this pandemic situation, which can be addressed and treated with the use of appropriate physical training and proper dietary habits.

The incidence of obesity in children has increased worldwide after COVID-19, and WHO has rightly noticed the various consequences of this problem. In different parts of the world, obesity in children is considered as a more serious health concern than malnutrition [4]. It is usually associated with immune vulnerability, and is a risk factor for many acute and chronic health concerns [5]. A systematic review and meta-analysis on the COVID-19 pandemic disease says that children with obesity tend to have a higher risk of developing the disease, a longer intensive care unit (ICU) stay, and more health problems than those who do not [6]. Therefore, it is not only considered as a pandemic disease, but also a gateway to ill health and many associated health problems [7]. In children affected with obesity, COVID-19 infection could result in increased subclinical inflammation and angiotensin converting enzyme-2 (ACE-2) level, and reduced immune responses, resulting in associated cardiorespiratory problems [8]. Vamanu et al. observed that oxidative stress due to microbial strains directly affects the structure of the microbial pattern in the gut of children, which, in later stages, leads to dysbiosis. It is also noted that there is a direct relationship between the incidence of dysbiosis and childhood obesity [9].

During the current pandemic, children have been advised to stay at home, which leads to limited mobility and a more sedentary lifestyle, which hastens weight gain and diminishes the overall function of our body [10]. Physical inactivity also makes systemic changes in our body, such as changes in inflammatory reaction, altered muscle protein and fat levels, a decrease in mitochondrial function, changes in hormone levels, general weakness and muscle pain, and weight gain [11,12]. The changes in fat or lipid levels in our body result in high cholesterol levels, dyslipidemia, hypertension, type 2 diabetes, joint pain, childhood asthma, and nonalcoholic fatty liver disease (NAFLD) [13,14]. Sheahan et al. observed that a lack of mobility and increased sedentary activities are the root cause of weight gain in rat models [15]. Usually, obese children confirmed with COVID-19 infection have some systemic illness that might exacerbate the condition. Therefore, they are advised to perform regular physical training and consume a proper diet to prevent and treat the negative consequences [16,17].

Therefore, different obesity management and weight reduction protocols have been developed to control and prevent the health problems and socio-economic issues associated with obesity. Many parents and caretakers of children with obesity usually prefer and actively adopt non-pharmacological approaches such as physical training, diet management, mind and body practice, behavioral therapy, lifestyle changes, and personal counseling for obesity management [4,18,19]. It has also been found through previous studies that effective obesity management protocols in children should consists of “realistic weight loss goals, prevention of weight regain, and maintenance of a healthy weight once achieved” [4,18,20]. Moreover, the acceptance of obesity as a serious health problem in children post-COVID-19-infection has raised a challenge for clinicians and allied health professionals to frame a proper physical training protocol and optimal dietary plan to overcome these consequences. The management of this clinical condition has received very little attention, and there are no well-defined exercise protocols or dietary prescriptions for this special population; therefore, there is a need for an elaborative trial in this field. Hence, the aim of this trial was to investigate and compare the clinical and psychological effects of integrated physical training with a high-protein diet versus a low-protein-diet in community-dwelling post-COVID-19-infection children with obesity. The reports of this trial can deliver the latest evidence and proper guidelines for the prescription of physical training and an optimal dietary plan in children with obesity following COVID-19 infection.

## 2. Methods

### 2.1. Study Design

This study is a single-blinded, randomized, clinical control trial conducted at the Department of Physical Therapy, Prince Sattam bin Abdulaziz University, Al Kharj, Saudi Arabia from May 2020 to December 2021. The trial received acceptance from the Department Ethical Committee (DEC) of Prince Sattam bin Abdulaziz University with an approval number of RHPT/020/058. The DEC approved the informed consent form, treatment protocol, and the outcome measures used in the trial. The trial was executed in accordance with the ethical guidelines laid down by the Declaration of Helsinki. The study was registered retrospectively with the registration number NCT05336006 on 19 April 2022. The children underwent basic clinical measurements for the eligibility to participate in the study. The procedures of the study were explained to the children and their parents, and their parents were asked to fill the informed written consent form. A two-block simple random sampling method was used to randomize and allocate them into two groups through sealed envelopes. Group A was treated with high-intensity aerobic training with a high-protein diet (*n* = 38) and Group B (*n* = 38) was considered as a control group, and we asked them to do regular physical activities for a duration of eight weeks.

### 2.2. Participants

Participants for the study were screened by a pediatrician, and recruited from local and government schools in the Al Riyadh and Al Kharj region. Positively-diagnosed 4-week post-COVID-19 obese male children in the age group of 5–12 years were recruited for the study (only male children were included because of disparities in the physical and hormonal scores). A body mass index (BMI) between the 85th and 99th percentiles was considered as obesity, and included in the study [21]. Subjects with a history of physical training, taking medications, recent surgeries, fractures and joint problems in lower extremities, cardiac and respiratory problems, neurological issues, major psychiatric problems, any other systemic diseases, contraindications for physical training, or family with food restrictions were excluded from the study. Participants participated in the study on their own and have not been paid any incentives. Figure 1 depicts the methods and procedures of involving the study subjects in this trial.

### 2.3. Intervention

Subjects in Group A (*n* = 38) underwent high-intensity aerobic training for 8 weeks. A trained physiotherapist instructed the subjects about the protocols for executing the intervention under proper COVID-19 instructions. The training protocol followed by the subject was recorded in a treatment logbook by the parent, and was assessed every week by the study supervisor. Prior to the start of exercise training, basic tests, such as body temperature, blood pressure, oxygen saturation rate, heartbeat, and physical fitness, were recorded. If these scores exceeded the normal values, i.e., body temperature >38.0 °C, blood-pressure >150/100 mmHg, and heartbeat >90 beat per min or <50 beat per min, the subject was asked to refrain from training in that session.

The maximum heart rate (MHR) was used to measure the intensity of the given exercise. It was calculated by reducing the subject’s age from the number 220. High-intensity aerobic training (HAT) was given at 50 to 70 percent of the maximum heart rate. Every day, the intervention started with 10 min of warm-up exercise, consisting of stretching of the upper and lower limb muscles. Subsequent to stretching, the subjects were asked to do 30 min of HAT exercises, consisting of 20 min of treadmill (Axos, Kettler Runner, China) and 10 min of cycle ergometer (Axos, Kettler Runner, China) at 50 to 70% of the MHR. Lastly, 10 min of cool-down was performed [16].

Next, the participants in Group A were prescribed strength training exercises, with resistance depending upon each subject’s individual muscle assessment. The ideal resistance for each muscle group was measured based on 1 repetition maximum (1RM) principle. The important muscle groups in the spine, and upper and lower limb were trained. Each muscle group underwent training of 3 sets consisting of 10 repetitions in each set, with a relaxation of 5 min rest between sets. The weight was gradually increased as per the subject’s and therapist’s needs, and the exercise was performed for 4 days a week for 8 weeks. In addition to these physical training exercises, Group A received a high-protein diet in the range of 1.1–1.2 g/kg protein/ideal body weight/day (>1 g/kg aBW/d), as prescribed by a qualified nutritionist based on an individual assessment and preferences, which includes both animal and vegetable proteins with a 1:1 ratio (Figure 2). The children were taught about the health consequences of overeating and having junk food, and informed about good eating behavior. They were allowed to have a high consumption of fruit, vegetables, whole grains, legumes, nuts, seeds, olive oil, and minimally processed foods [17]. Group B was considered as the control group, and they were allowed to follow their regular physical activities and dietary patterns.

### 2.4. Outcome Measures

The primary outcome measure was BMI percentile, which was measured at the 6-month period. The secondary outcome measures were muscle cross-sectional area and biochemical markers, which were collected at baseline, at 8 weeks, and at 6 months.

#### 2.4.1. Body Mass Index (BMI)

For children, age-adjusted BMI percentile (BMI %) was calculated, which is a reliable and valid measurement to measure the stage of obesity [22].

#### 2.4.2. Muscle

Cross-sectional area—CSA: Muscle CSA is measured with a magnetic resonance imaging (MRI) scan (Philips Ingenia, 1.5 TS, Cambridge, MA, USA), which is an expensive measurement. The CSA of three major muscles, i.e., halfway at arm-biceps, thigh-quadriceps, and calf muscles, were measured and included for analysis [23].

#### 2.4.3. Biochemical Analysis

Fasting (less than 12 h) venous blood samples were collected from all the participants, and centrifugation of the specimen was performed. Serum and plasma were separated and stored immediately at −80 °C. The biochemical markers adiponectin, leptin, TNF-α, and IL-6 levels were measured with an ELISA kit (Adiponectin (human) ELISA kit, Adipogen life sciences, San Diego, CA, USA) [24].

### 2.5. Sample Size

The sample size for the study was decided based on a report from the study by Boutelle et al. [22]. Thirty-six subjects in each group were calculated by assuming 20% mean improvement in BMI% with a standard deviation of 2. By accepting a 10% drop of subjects at 6-months follow-up, two subjects were added in each group. Hence, in total, 76 subjects were selected and divided into two groups (*n* = 38). The sample size was calculated with the statistical power at 80% with a significance α level of 0.05. The whole analysis was done using G*Power—version 3.1.9.7 (UCLA, Los Angeles, CA, USA).

### 2.6. Statistical Analysis

Subjects’ personal and anthropometric measurements were calculated through the Kolmogorov—Smirnov test for testing homogeneity, and the data were represented in tabular form. The measurements were taken before intervention, after intervention at 8 weeks, and at 6-months follow-up. The data were shown as mean and standard deviation with a 95% confidence interval (CI) with the upper and lower limit. The time and group (3 × 2) multiple analysis of variance (MANOVA) of primary and secondary variables is reported between Group A and Group B at various intervals. An independent *t*-test was used to calculate inter-group effects, and repeated measures (rANOVA) were used to calculate the intra-group effects. IBM SPSS—online version 20 (Chicago, IL, USA) was used to perform all the statistical tests, and the α level was set at 0.05.

## 3. Results

Initially, a pediatrician screened one-hundred and twenty-six subjects for the eligibility to be recruited for the study. Among these, fifteen had suffered from some systemic illness, fifteen had joint problems, eight subjects had undergone some type of surgery, and twelve subjects did not consent to be involved in the trial. Seventy-six (*N* = 76) subjects were selected and allocated into study groups. The whole study trial and the tests were conducted by assuming the intention-to-treat principle method. Two participants in Group A and two participants in Group B did not complete the follow-up measurement at 6 months due to some personal inconveniences, which was shown in Figure 1. The Kolmogorov–Smirnov test was used to find the subject’s personal and anthropometric traits, which show no statistical difference between the groups (*p* > 0.05). This test reports the homogenous distribution of the subjects between the groups, and leads to further statistical analysis. The other physiologic measures, such as VO_2_ max (maximum oxygen volume) and heartbeat, were also checked for identifying the ability to participate in physical training; these data also show no statistical difference between the groups (*p* > 0.05). The detailed description of demographic and clinical characteristics was presented as mean and standard deviation, and is presented in Table 1.

The time and group (3 × 2) MANOVA of the primary variable (body mass index (BMI)—percentile) reports a statistically significant difference (*p* < 0.001) between Group A and Group B at baseline, at 8 weeks, and at 6 months. The pre-intervention measure of BMI did not report a significant change (*p* > 0.05), but at 8 weeks, there was a 2.8 (CI 95% 1.17 to 4.42) improvement in Group A compared to Group B. Similar effects were observed at 6-months follow-up measurement, which shows a 6.3 (CI 95% 4.71 to 7.88) greater effect (*p* < 0.001) in Group A than Group B, which is shown in Table 2. The effect size of body mass index (d = 1.82) shows a larger effect in Group A than Group B.

The time and group (3 × 2) MANOVA of the secondary variable, muscle CSA–MRI (arm, thigh, and calf), reports statistically significant difference (*p* < 0.001) between Group A and Group B at baseline, at 8 weeks, and at 6 months. The pre-intervention measure of muscle CSA values did not report any statistical change (*p* > 0.05), but at 8 weeks, mid-arm CSA (10.8) (CI 95% 8.51 to 13.08), mid-thigh CSA (7.7) (CI 95% 5.23 to 10.16), and mid-calf CSA (7.0) (CI 95% 4.95 to 9.04) showed improved scores (*p* < 0.001) in Group A compared to Group B. The same effects were noted at 6-months follow-up measurement in mid-arm CSA (17) (CI 95% 14.70 to 19.29), mid-thigh CSA (13.10) (CI 95% 10.60 to 15.59), and mid-calf CSA (11.3) (CI 95% 9.30 to 13.29), which showed greater improvement (*p* < 0.001) in Group A than Group B, which is shown in Table 2. The effect size of mid-arm CSA (d = 3.40), mid-thigh CSA (d = 2.40), and mid-calf CSA (d = 1.67) shows a larger effect in Group A than Group B. The visual representation in Figure 3 also shows more improvements in all variables in Group A than in Group B.

The time and group (3 × 2) MANOVA of the secondary variable, biochemical markers (adiponectin, leptin, TNF-α, and IL-6), reports a statistically significant difference (*p* < 0.001) between group A and group B at baseline, at 8 weeks, and at 6 months. The pre-intervention measure of biochemical markers did not report any statistical change (*p* > 0.05), but at 8 weeks, adiponectin (−1.3) (CI 95% −1.55 to −1.10), leptin (3.01) (CI 95% 2.89 to 3.12), TNF-α (0.2) (CI 95% 0.12 to 0.27), and IL-6 (0.09) (CI 95% 0.06 to 0.11) showed improved scores (*p* < 0.001) in Group A compared to Group B. The same effects were noted at 6-months follow-up measurement in adiponectin (−1.9) (CI 95% −2.13 to −1.66), leptin (5.64) (CI 95% 5.50 to 5.77), TNF-α (0.5) (CI 95% 0.42 to 0.57), and IL-6 (0.21) (CI 95% 0.18 to 0.23), which showed greater improvement (*p* < 0.001) in Group A than Group B, which is shown in Table 2. The effect size of adiponectin (d = 3.80), leptin (d = 19.72), TNF-α (d = 3.33), and IL-6 (d = 4.66) shows a larger effect in Group A than in Group B. The visual representation in Figure 2 also shows more improvements in all variables in Group A than in Group B.

## 4. Discussion

This study was conducted to find the clinical and biochemical effects of high-intensity aerobic training (HAT) with a high-protein diet in children with obesity (CO) following COVID-19 infection. The medical and associated allied health professionals believed that proper physical training and a balanced food pattern would be helpful to treat and control obesity during childhood in this pandemic situation. Guan et al. observed that obesity in children following COVID-19 infection could be due to the long stay at home because of social isolation and closure of schools due to government lockdown. They advised regular physical activities, and recommended a high-protein diet for controlling obesity in children [25]. The data of this study showed that high-intensity aerobic training with a high-protein diet has better effects than regular physical activities and food patterns in clinical and biochemical effects in children with obesity following COVID-19 infection.

High-intensity aerobic training activities are safe exercise approaches, which activate the entire body and maintain the muscles’ physiologic properties. Atlantis et al. conducted a systematic review on obesity in childhood and found that a 0.4% (0.1–0.7%) reduction in body fat percentage occurred with HAT exercises performed for 155–180 min weekly with an energy consumption of 1000 to 2000 kcal per week [26]. HAT generally induces energy activation, maintains the physical wellbeing, and controls obesity in childhood [27]. It was found that high-intensity aerobic training exercise prevents the adverse health consequences of obesity. These changes occur by altering the inflammatory reaction, mitochondrial activity, and insulin response. HAT enhances the peripheral circulation and increases the local blood flow to the cells, which enhances overall muscle endurance. This type of training activates the muscles’ oxidative capacity and improves the function of mitochondria in the cell [28]. Moreover, a balanced diet, which consists of optimum protein supplement, is required to maintain muscle properties in children with obesity. Goldthorpe et al. conducted a survey in Europe, and noticed that around 35% of children were not consuming the estimated average requirement (EAR) of protein per day (0.7 g/kg body weight/day). This imbalance in protein intake and protein demand may lead to alteration in muscle size, muscle strength, and overall performance, which eventually affects quality of life [29].

Our study reported significant differences in the primary outcome variable, i.e., BMI percentile, between Group A and Group B. Tripathi et al. observed increased BMI percentile and body fat percentage in children with obesity [30]. Generally, children with obesity develop anabolic resistance, which resists the action of protein synthesis, and diminishes the role of metabolism [31]. Therefore, there is an increase in the demand of protein intake by approximately 1.0 to 1.2 g/kg body weight/day in children with obesity undergoing physical training, which was recommended by Dideriksen et al. [32]. According to Koopman R et al., consuming a high-protein diet (Group A) increases the secretion of amino acids (AA) from the gut, reduces the post-meal requirement of AA and post-meal introduction of AA into muscle, decreases the anabolic resistance, and reduces the effort of digestive system [33]. The within-group analysis through repeated measures ANOVA reports substantial changes in BMI percentile in Group A compared to Group B. In addition, the minimum clinically important difference (MCID) score in BMI percentile (6.3) between the two groups shows a greater effect in Group A than in Group B.

Our study reported decreased muscle CSA of the arm, thigh, and calf region in Group A compared to Group B, which was in agreement with Kim et al. [17]. Roth et al. observed that the sequence of doing exercise is an important factor (for instance, strength training followed by aerobic training) in changing the muscle tone and size [34]. In our trial, in the first phase, aerobic training was done, and in the second phase, strengthening exercise was done, which could be the reason for little changes in muscle cross-sectional area. The real physiological and biochemical changes behind the sequence of performing exercise was not clearly investigated. In addition, it was found that the minimal clinical changes in muscle CSA of the arm, thigh, and calf region may be due to either exercise training parameters or the type of dietary supplements.

The study also found that HAT with high-protein diet positively enhanced the biochemical attributes in children with obesity following COVID-19 infection. Obesity in childhood induces the secretion of adipocytokines such as leptin, TNF-α, and IL-6, and slows the secretion of adiponectin, which causes insulin resistance. A lower level of adiponectin is an indicator for a high risk of insulin resistance. Adiponectin is a fat-derived hormone that appears to play a crucial role in protecting against insulin resistance/diabetes and atherosclerosis. It reduces the action of adipocytokines such as leptin, TNF-α, and IL-6, as well as oxidative stress, which leads to an improvement of insulin resistance. It is associated with the lipoprotein metabolism, and increases high-density lipoprotein (HDL) and decreases triglyceride (TG) in the body. Adiponectin increases ATP-binding cassette transporter A1 and lipoprotein lipase (LPL), and decreases hepatic lipase, which may elevate HDL. These findings suggest that high levels of circulating adiponectin can protect against weight gain [35]. Wu S et al. stated that the quality of protein and optimal food processing conditions are important, and have a positive effect on gut microbiota and dysbiosis [36]. Our study reported increased adiponectin levels in Group A, which was supported by Kondo et al. [37] and Bluher et al. [38], but the opposite was reported by Nassis et al. [39]. These reports suggested that children have higher metabolic flexibility towards adiponectin secretion in response to high-intensity aerobic training. Our study also reported decreases in the levels of leptin, TNF-α, and IL-6 after intervention, which was in agreement with a previous study by Esposito et al. [40]. It was also observed by Kim ES et al. that adiponectin regulates other biomarkers, such as leptin, TNF-α, and IL-6 levels, in children with obesity. High-intensity aerobic training with a high-protein diet reduces the levels of fat accumulation in our body which may cause an increase in adiponectin level. Positive changes in biomarkers directly deplete the fat deposition in children with obesity, which slows down the disease process. The difference in outcome variables between the groups is due to their variations in protein consumption. Moreover, the physiology of the body reaction to a high-protein diet is greater in young adults than those of an older age [41]. These clinical and biochemical changes improve the overall wellbeing of the children with obesity following COVID-19 infection.

The authors faced some limitations and difficulties while executing the study. First, subjects in both groups underwent the physical training programs at the same venue and at the same time, which may affect the masking of participants, because they may share the exercise intervention and study protocols. Second, this study did not include female participants, which may have provided the effects of intervention between male and female participants. Third, as it is a preliminary study, the other pro-inflammatory cytokines such as resistin and visfatin were not included in the biochemical analysis. Fourth, the authors did not find the association between the clinical and biochemical traits after the study. Finally, the study did not include a healthy control group, which may have presented the actual effects of HAT with a high-protein diet. Hence, further studies can be conducted by masking the participants and including a healthy control group. Future studies are also recommended to find the association between the clinical and biochemical traits after the study intervention.

## 5. Conclusions

Overall, this trial found that high-intensity aerobic training with a high-protein diet decreased the BMI percentile and muscle mass (arm, thigh, and calf), and positively altered the biochemical variables in children with obesity following COVID-19 infection. This trial provides a greater insight in the field of pediatric rehabilitation for medical and allied health workers to prevent or treat children with obesity.

## Figures and Tables

**Figure 1 ijerph-19-07194-f001:**
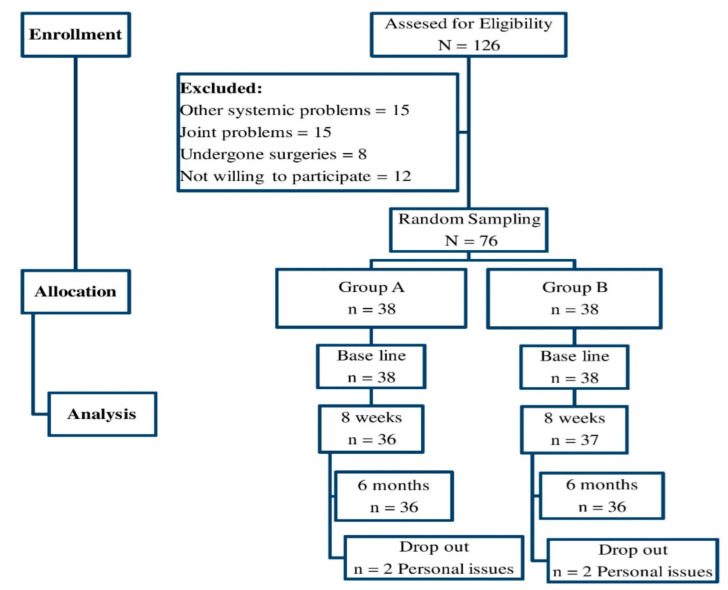
Flow chart showing the study details.

**Figure 2 ijerph-19-07194-f002:**
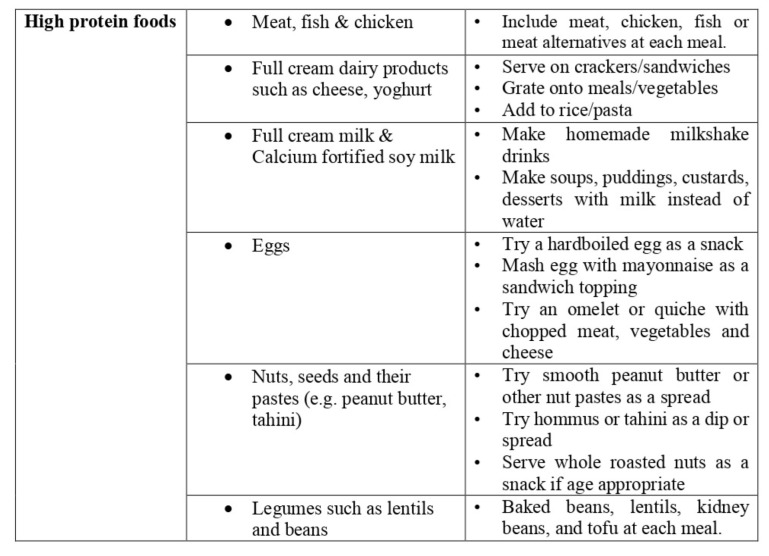
Recommended high-protein diet chart.

**Figure 3 ijerph-19-07194-f003:**
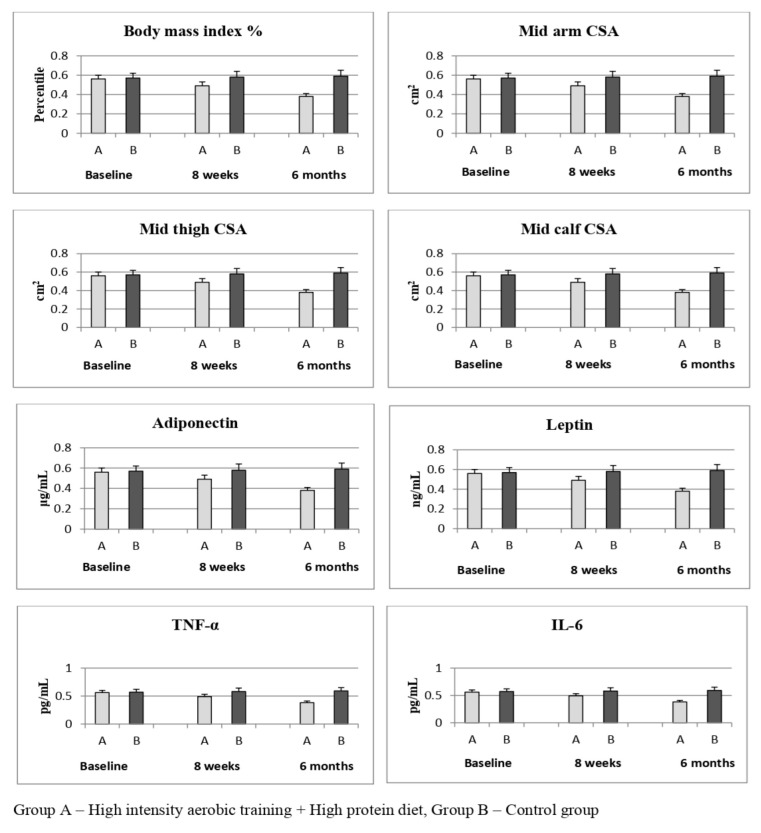
Pre and post primary and secondary outcome measures of Group A and B.

**Table 1 ijerph-19-07194-t001:** Demographic and clinical characteristics of Group A and B.

Sr. No	Variable	Group A	Group B	*p*-Value
1	Age (y)	10.12 ± 1.2	10.56 ± 1.4	0.145
2	Height (m)	1.28 ± 0.12	1.30 ± 0.10	0.432
3	Weight (kg)	43.8 ± 4.3	44.2 ± 4.8	0.703
4	VO_2_peak (mL/kg/min)	40.3 ± 3.9	39.2 ± 3.6	0.205
5	HR (beats/min)	76.8 ± 5.5	78.2 ± 4.9	0.245

**Table 2 ijerph-19-07194-t002:** Pre and post primary and secondary outcome measures of Group A and B.

Sr. No	Variable	Duration	Group A	Group B	*p*-Value
1	Body mass index % (Percentile)	Base line	98.8 ± 3.5	99.1 ± 3.7	0.717
		8 weeks	96.5 ± 3.4	99.3 ± 3.7	0.001 *
		6 months	93.2 ± 3.1	99.5 ± 3.8	0.001 *
		*p*-value	0.001 *	0.896	
2	Muscle quantity—CSA (MRI—Mid-arm: cm^2^)	Base line	59.8 ± 5.1	58.9 ± 5.3	0.453
		8 weeks	48.2 ± 4.7	59.0 ± 5.3	0.001
		6 months	42.2 ± 4.5	59.2 ± 5.5	0.119
		*p*-value	0.001 *	0.969	
3	Muscle quantity—CSA (MRI—Mid-thigh: cm^2^)	Base line	67.2 ± 5.5	67.1 ± 5.6	0.937
		8 weeks	61.2 ± 5.3	68.9 ± 5.5	0.001 *
		6 months	56.5 ± 5.2	69.6 ± 5.7	0.001 *
		*p*-value	0.001 *	0.138	
4	Muscle quantity—CSA (MRI—Mid-calf: cm^2^)	Base line	59.5 ± 4.6	58.9 ± 4.5	0.567
		8 weeks	52.5 ± 4.1	59.5 ± 4.8	0.001 *
		6 months	48.3 ± 3.9	59.6 ± 4.8	0.001 *
		*p*-value	0.001 *	0.782	
5	Adiponectin (μg/mL)	Base line	7.52 ± 0.4	7.45 ± 0.5	0.502
		8 weeks	8.74 ± 0.5	7.41 ± 0.5	0.001 *
		6 months	9.28 ± 0.6	7.38 ± 0.4	0.001 *
		*p*-value	0.001 *	0.808	
6	Leptin (ng/mL)	Base line	12.25 ± 0.2	12.15 ± 0.3	0.091
		8 weeks	9.33 ± 0.2	12.34 ± 0.3	0.001 *
		6 months	7.25 ± 0.1	12.18 ± 0.4	0.001 *
		*p*-value	0.001 *	0.833	
7	TNF-α (pg/mL)	Base line	1.7 ± 0.1	1.7 ± 0.2	1.000
		8 weeks	1.5 ± 0.1	1.7 ± 0.2	0.001 *
		6 months	1.3 ± 0.1	1.8 ± 0.2	0.001 *
		*p*-value	0.001 *	0.046	
8	IL-6 (pg/mL)	Base line	0.56 ± 0.04	0.57 ± 0.05	0.338
		8 weeks	0.49 ± 0.04	0.58 ± 0.06	0.001 *
		6 months	0.38 ± 0.03	0.59 ± 0.06	0.001 *
		*p*-value	0.001 *	0.312	

* Significant, Group A—high-intensity aerobic training + high-protein diet, Group B—control group, MRI—magnetic resonance imaging, CSA—cross-sectional area.

## Data Availability

Data are not publicly available, but can be obtained from the corresponding author on request.

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
