# Peer review of "Clinical (BMI and MRI) and Biochemical (Adiponectin, Leptin, TNF-α, and IL-6) Effects of High-Intensity Aerobic Training with High-Protein Diet in Children with Obesity Following COVID-19 Infection"

_ijerph, 2022, doi:10.3390/ijerph19127194_

Round 1

Reviewer 1 Report

The article is written at a high level with benefits for the scientific community. However, I would like to comment on some of  more important points in this article:

In the title you specify: "children with obesity following COVID-19 infection" If I understood the article correctly, they are obese children whose obesity has been exacerbated by the pandemic and isolation due to COVID-19. However, the name evokes that these are all children after overcoming COVID-19, which is probably not true, as the article no longer focuses on the parameters that are monitored as post-COVID-19.

The second thing that bothers me is the lack of discussion about the effect of adiponectins on adipocytokines such as leptin, TNF-α and IL-6. Their effect on metabolism of lipids, sacharides.... An explanation of why you just chose these and not eg vistafin or resistin, etc.

Author Response

Reviewer 1

Reviewer comment 1: The article is written at a high level with benefits for the scientific community. However, I would like to comment on some of more important points in this article:

Author response: Thank you so much for the constructive and positive comments. Indeed, the comments was much helpful to improve the scientific quality and technical soundness of the article. As per your suggestion the required changes have been done in the manuscript.

Reviewer comment 2: In the title you specify: "children with obesity following COVID-19 infection" If I understood the article correctly, they are obese children whose obesity has been exacerbated by the pandemic and isolation due to COVID-19. However, the name evokes that these are all children after overcoming COVID-19, which is probably not true, as the article no longer focuses on the parameters that are monitored as post-COVID-19.

Author response: Thank you for the comments and suggestion. The title of the study also modified like post COVID-19 infection.

As per your suggestion the required modifications have been done in the introduction and methods section. The study participants are the obese children and their symptoms are exacerbated due to COVID-19 infection.

We are sorry for this mistake, and in the introduction and methods section the details (post COVID-19) about the study participants were added. 

Reviewer comment 3: The second thing that bothers me is the lack of discussion about the effect of adiponectins on adipocytokines such as leptin, TNF-α and IL-6. Their effect on metabolism of lipids, sacharides.... An explanation of why you just chose these and not eg vistafin or resistin, etc.

Author response: As per your suggestion the required modifications have been done in the discussion section.

The effect of adiponectin on adipocytokines such as leptin, TNF-α and IL-6 has been mentioned.

The effect of adiponectin on metabolism of lipids and saccharides has been explained with reference.

As, it is a preliminary study the other pro-inflammatory cytokines like resistin and visfatin were not included in the biochemical analysis and it was added in the limitation of the study.

Once again, we are appreciating your efforts and we are happy to address any further queries and comments in further revisions. Thank you.

Reviewer 2 Report

The work turns out to be interesting, even if some points need to be revised.

The layout must be reviewed according to the newspaper's rules, including the abstract. Furthermore, the references in brackets must be placed before the point, because so it seems that they refer to the concept that comes after.

In the keywords replace biochemical with adipocytokines.

The age of the participants is in a very wide range (5-12 years) this can affect the results obtained, also given a hormonal involvement?

Is there a difference in adiponectin levels between males and females?

I suggest making a table with the characteristics of the diet shown.

Are the differences statistically significant in Figure 2? It is not clear.

I suggest rewriting the materials and methods in a clearer form.

Author Response

Reviewer 2

Reviewer comment 1: The work turns out to be interesting, even if some points need to be revised.

Author response: Thank you so much for the constructive and positive comments. Indeed, the comments was much helpful to improve the scientific quality technical soundness of the article. As per your suggestion the required changes have been done in the manuscript.

Reviewer comment 2: The layout must be reviewed according to the newspaper's rules, including the abstract. Furthermore, the references in brackets must be placed before the point, because so it seems that they refer to the concept that comes after.

Author response: As per your suggestion the abstract and the whole manuscript is reviewed and modified as per the journal guidelines and also the square brackets are kept before the full stop.

Reviewer comment 3: In the keywords replace biochemical with adipocytokines.

Author response: As per your suggestion the keyword biochemical is replaced with the word adipocytokines.

Reviewer comment 4: The age of the participants is in a very wide range (5-12 years) this can affect the results obtained, also given a hormonal involvement?

Author response: Thank you for raising this comment, which will provide the clear information about the age range of participants. Though the age range is between 5-12 years the statistical analysis of the age range between the two groups shows no significant difference, which is shown in the table 1. Hence the wide age range such as 5 – 12 years will not affect the results obtained.

Reviewer comment 5: Is there a difference in adiponectin levels between males and females?

Author response: The study involves only male obese children because of disparities in the physical and hormonal scores, which may affect the final reports of the study. Also this is included as one of the limitation of the study.

Reviewer comment 6: I suggest making a table with the characteristics of the diet shown.

Author response: The diet chart was prescribed by a qualified nutritionist based on an individual assessment and preferences. As per your suggestion the recommended high protein diet chart has been added as figure 1.

Reviewer comment 7: Are the differences statistically significant in Figure 2? It is not clear.

Author response: The scores in the figure 2 are statistically significant between the groups. Due to space constraint the figure 2 is compressed. The figure is in jpg format, so that it can be stretched during publication.

Reviewer comment 8: I suggest rewriting the materials and methods in a clearer form.

Author response: Thank you for the suggestion and as per your comments, the materials and methods section is overviewed fully and required changes have been done.

Once again, we are appreciating your efforts and we are happy to address any further queries and comments in further revisions. Thank you.

Reviewer 3 Report

The content of this article is fully consistent with its title. Presented a problem developed on the basis ofcurrent knowledge. Using appropriate research methods. The scope of the analysis and interpretation of test results is correct. Results and conclusions represent a contribution to the development of this field of knowledge. The subject of this article is important. Graphic illustration of the text is correct.

Throughout the manuscript, citations in the text should be corrected as they do not conform to journal requirements.

In chapter 2.3. please describe what type of protein the children consumed. Please state whether it was animal or vegetable protein. What was the animal to vegetable protein ratio.

The problem of childhood obesity, COVID-19 and immunity is very complicated. In the introduction and discussion, the authors should take a closer look at the problem of obesity and describe the problem of dysbiosis and the impact of a high-protein diet on the intestinal microbiota and the consequences of such a diet.

Author Response

Reviewer 3

Reviewer comment 1: The content of this article is fully consistent with its title. Presented a problem developed on the basis of current knowledge. Using appropriate research methods. The scope of the analysis and interpretation of test results is correct. Results and conclusions represent a contribution to the development of this field of knowledge. The subject of this article is important. Graphic illustration of the text is correct.

Author response: Thank you so much for the constructive and positive comments. Indeed, the comments was much helpful to improve the scientific quality technical soundness of the article. As per your suggestion the required changes have been done in the manuscript.

Reviewer comment 2: Throughout the manuscript, citations in the text should be corrected as they do not conform to journal requirements.

Author response: As per your suggestion the abstract and the whole manuscript is reviewed and modified as per the journal guidelines and also the square brackets are kept before the full stop.

Reviewer comment 3: In chapter 2.3. please describe what type of protein the children consumed. Please state whether it was animal or vegetable protein. What was the animal to vegetable protein ratio.

Author response: As per your suggestion the information regarding the type of protein the children consumed and it’s ratio between the proteins.

Reviewer comment 4: The problem of childhood obesity, COVID-19 and immunity is very complicated. In the introduction and discussion, the authors should take a closer look at the problem of obesity and describe the problem of dysbiosis and the impact of a high-protein diet on the intestinal microbiota and the consequences of such a diet.

Author response: As per your suggestion, the required changes have been done in the introduction and discussion part and new references have been added (reference number 9 and 39).

Once again, we are appreciating your efforts and we are happy to address any further queries and comments in further revisions. Thank you.

Round 2

Reviewer 2 Report

the manuscript is ok for publication.